

# Availability of Nanopore sequences in the genome taxonomy for *Vibrionaceae* systematics: Rumoiensis clade species as a test case

Mami Tanaka[1], Sayaka Mino[1], Yoshitoshi Ogura[2], Tetsuya Hayashi[2] and Tomoo Sawabe[1]

[1] Laboratory of Microbiology, Faculty of Fisheries, Hokkaido University, Hakodate, Japan
[2] Department of Bacteriology, Faculty of Medical Sciences, Kyushu University, Fukuoka, Japan

## ABSTRACT

Whole genome sequence comparisons have become essential for establishing a robust scheme in bacterial taxonomy. To generalize this genome-based taxonomy, fast, reliable, and cost-effective genome sequencing methodologies are required. MinION, the palm-sized sequencer from Oxford Nanopore Technologies, enables rapid sequencing of bacterial genomes using minimal laboratory resources. Here we tested the ability of Nanopore sequences for the genome-based taxonomy of *Vibrionaceae* and compared Nanopore-only assemblies to complete genomes of five Rumoiensis clade species: *Vibrio aphrogenes, V. algivorus, V. casei, V. litoralis,* and *V. rumoiensis*. Comparison of overall genome relatedness indices (OGRI) and multilocus sequence analysis (MLSA) based on Nanopore-only assembly and Illumina or hybrid assemblies revealed that errors in Nanopore-only assembly do not influence average nucleotide identity (ANI), *in silico* DNA-DNA hybridization (DDH), G+C content, or MLSA tree topology in *Vibrionaceae*. Our results show that the genome sequences from Nanopore-based approach can be used for rapid species identification based on the OGRI and MLSA.

# INTRODUCTION

In the last decade, high-throughput DNA sequencing technologies has dramatically expanded the genetic information of all domains of life and viruses, and has contributed to the generation of 2,408 *Vibrionaceae* genomes deposited in the National Center for Biotechnology Information (NCBI) database (http://www.ncbi.nlm.nih.gov/genome/; December 22, 2017). Whole genome sequencing has become a routine experiment in bacterial taxonomy as the recently emerging "Genome-based Taxonomy" approach provides reproducible, reliable, and highly informative data for phylogenetic inference, differentiating species without the need for specialized skills (*Chun et al., 2018*). Traditionally, genomic coherence between strains has been determined through DDH experiments, and a cutoff value of 70% DDH for circumscribing species has been recognized as "the golden standard" in prokaryotes taxonomy (*Rosselló-Móra & Amann, 2015*; *Chun et al., 2018*). OGRI (*Chun & Rainey, 2014*) such as ANI (*Konstantinidis & Tiedje, 2005*;

Corresponding author
Tomoo Sawabe,
sawabe@fish.hokudai.ac.jp

*Goris et al., 2007*) and genome-to-genome distance (GGD) (*Auch et al., 2010*) have been utilized to replace traditional DDH, with 95–96% ANI and 70% *in silico* DDH from GGD corresponding to 70% experimental DDH (*Rosselló-Móra & Amann, 2015*; *Chun et al., 2018*). However, whole genome sequencing, particularly methods using Illumina sequencers which are currently the most widely used worldwide, usually require larger facilities and costly maintenance, which could make species identification impractical in smaller laboratories.

Oxford Nanopore Technologies MinION is a palm-sized sequencer, and the ability to produce long reads in real time made Nanopore sequencing an attractive option for genomics. MinION could typically generate 5–10 Gb DNA in a single run (*Magi et al., 2017*), dramatically reducing the sequencing cost for bacterial genomes. Its accessibility in terms of cost and minimal equipment needs enables rapid acquisition of whole genome data even in non-specialized laboratories. A shortcoming is its relatively high error rate. Nanopore-only assembly, in contrast to hybrid approach combining Illumina reads, still faces challenges regarding its accuracy, which could limit the use of Nanopore sequencing technology (*Wick et al., 2017a*).

As *Vibrionaceae* are metabolically and genetically diverse, this group of bacteria have always been at the forefront of bacterial taxonomy being tested for new methodologies (*Sawabe et al., 2013*). Among 24 clades in *Vibrionaceae*, the Rumoiensis clade consists of species with diverse ecophysiology, but at the same time the clade is phylogenetically robust (*Tanaka et al., 2017*). As there are currently only five species in the clade, we were able to finish the genomes of all type strains of the Rumoiensis clade. The aim of this study is to test the availability of Nanopore sequencing in creating a rapid, reliable, and ultimately automatic identification scheme for *Vibrionaceae*. Using the Rumoiensis clade species as the test case, we compared OGRI and MLSA topology based on data obtained from Nanopore-only assembly and hybrid assembly. Every OGRI determined in this study shows that Nanopore sequencing is applicable for the genome-based taxonomy in *Vibrionaceae*.

## MATERIALS & METHODS

### Bacterial strains

*V. aphrogenes* CA-1004[T] (=JCM 31643[T]), *V. algivorus* NBRC 111146[T], *V. casei* DSM 22364[T], *V. litoralis* DSM 17657[T], and *V. rumoiensis* FERM P-14531[T] used in this study were cultured on ZoBell 2216E agar unless otherwise indicated.

### DNA extraction

Cells were pre-cultured in ZoBell 2216E broth using natural seawater at 25 °C for 24 h with shaking at 120 rpm. Genomic DNA was extracted using the Wizard® Genomic DNA Purification Kit (Promega Corporation, Madison, WI, USA) following the protocol for Gram negative bacteria with minor modifications. For *V. aphrogenes*, genomic DNA was also extracted using the NucleoSpin® Tissue (MACHEREY-NAGEL GmbH & Co. KG, Düren, Germany) with a protocol for Gram negative bacteria. DNA concentration was measured using the Quantus™ Fluorometer with the QuantiFluor® ONE dsDNA System (Promega Corporation, Madison, WI, USA). Purity was evaluated by measuring

A260/A280 and A260/A230 using the Eppendorf BioSpectrometer® kinetic (Eppendorf AG, Hamburg Germany) with the Eppendorf μCuvette® G1.0 and the dsDNA 1 mm protocol. To check DNA size after the extraction, 100 ng of genomic DNA was applied to 0.8% agarose gel electrophoresis and DNA bands were visualized with the Printgraph2M (ATTO CORPORATION, Tokyo, Japan).

## Whole genome sequencing

Sequencing library for Nanopore sequencing was prepared using the Rapid Barcoding Kit (SQK-RBK001) (Oxford Nanopore Technologies, Oxford, UK) following the protocol supplied by the manufacturer. The library was then loaded to MinION set with a flow cell FLO-MIN106 R9 version (Oxford Nanopore Technologies, Oxford, UK) and the sequencing run was performed under MinKNOW version 1.7.14.

PacBio sequencing of *V. aphrogenes* genome was performed at The Center of Medical Innovation and Translational Research, Graduate School of Medicine, Osaka University. The library was prepared using the SMRTbell template prep kit 1.0 and the DNA polymerase binding kit P6 version 2. Sequencing was performed using a single SMRT® cell with the PacBio® RS II System (Pacific Biosciences, Menlo Park, CA, USA).

Illumina paired-end reads (PE) for *V. aphrogenes*, *V. algivorus*, *V. casei*, and *V. rumoiensis*, and mate-pair (MP) reads for *V. aphrogenes* were previously obtained (*Tanaka et al., 2017*) using the MiSeq platform. Briefly, PE libraries were prepared using the Nextera XT DNA Library Preparation Kit for *V. aphrogenes* and the TruSeq PCR-Free Kit for *V. algivorus*, *V. casei*, and *V. rumoiensis*. A 8 kb MP library for *V. aphrogenes* was constructed using the Nextera Mate Pair Sample Preparation Kit. Illumina HiSeq PE reads for *V. litoralis* were obtained from NCBI SRA under accession SRR896479.

## Reads processing and *de novo* assembly

Fast5s from Nanopore sequencing were basecalled with ONT Albacore Sequencing Pipeline Software version 2.0.2 and reads passing the internal test were used for subsequent analysis. Porechop 0.2.2 (https://github.com/rrwick/Porechop) was used for debarcoding and adaptor trimming. Illumina PE reads were processed with platanus_trim in Platanus (*Kajitani et al., 2014*) to trim adaptor sequences. Illumina-only assembly was performed using Platanus (*Kajitani et al., 2014*) optimized for bacterial genomes.

Nanopore reads were assembled using Canu 1.6 (*Koren et al., 2017*) with genomeSize = 3.5 m. For Nanopore-only assembly, output contigs were polished using Nanopolish version 0.8.1 (https://github.com/jts/nanopolish). Contigs were manually circularized by confirming the overlap regions at the ends of each contig. Hybrid assembly using Nanopore and Illumina PE reads was performed using Unicycler v0.4.2 (*Wick et al., 2017b*) with minor modifications to match the starting positions of *Vibrio* genomes. For *V. rumoiensis*, contigs from Canu were manually closed based on the assembly graph with Bandage version 0.8.1 (*Wick et al., 2015*), and circular contigs were polished with Illumina PE reads using Pilon version 1.22 (*Walker et al., 2014*). SeqKit version 0.7.1 (*Shen et al., 2016*) was used for FASTA/FASTQ handlings. PacBio sequences were assembled using HGAP3 protocol in SMRT® Analysis version 2.3.0 (*Chin et al., 2013*). Polished contigs
**Table 1** **Assembly results of the *V. aphrogenes* genome using reads from different platforms.** Resulting assemblies were evaluated using QUAST v4.5 (*Gurevich et al., 2013*).

| Assembly | PacBio +Illumina | PacBio | Nanopore +Illumina | Nanopore | MP+PE | PE |
|---|---|---|---|---|---|---|
| Assembler | HGAP +Pilon | HGAP | Unicycler | Canu + Nanopolish | Platanus (MP) | Plataus (PE) |
| Number of contigs (≥ 0 bp) | 2 | 2 | 2 | 2 | 40 | 51 |
| Total length (bp) (≥ 0 bp) | 3,375,422 | 3,375,390 | 3,375,144 | 3,371,144 | 3,371,804 | 3,333,369 |
| Number of contigs (≥ 1 kb) | 2 | 2 | 2 | 2 | 2 | 23 |
| Total length (bp) (≥ 1 kb) | 3,375,422 | 3,375,390 | 3,375,144 | 3,371,144 | 3,360,281 | 3,322,746 |
| Genome fraction (%) | 100 | 100 | 99.987 | 100 | 99.512 | 98.433 |
| Number of N per kb | 0 | 0 | 0 | 0 | 0.3458 | 0.0767 |
| Mismatches per kb | 0 | 0.0003 | 0.0033 | 0.0240 | 0.0122 | 0.0069 |
| Indels per kb | 0 | 0.0095 | 0.0124 | 1.3569 | 0.0158 | 0.0160 |

were manually circularized. Assemblies were further polished with Illumina PE reads using Pilon version 1.22 (*Walker et al., 2014*).

## Assembly statistics and overall genome relatedness indices

General assembly statistics including total length, DNA G+C content, and indels/mismatches against reference genomes were calculated using QUAST v4.5 (*Gurevich et al., 2013*). ANI values were determined using Orthologous Average Nucleotide Identity Tool version 1.3 (*Lee et al., 2016*). *In silico* DDH values were estimated using Genome-to-Genome Distance Calculator (GGDC) 2.1 (*Nelder & Wedderburn, 1972*; *Meier-Kolthoff et al., 2013*).

## Multilocus sequence analysis and tree comparison

Eight protein coding genes (*gapA*, *gyrB*, *ftsZ*, *mreB*, *pyrH*, *recA*, *rpoA*, and *topA*) used for *Vibrionaceae* MLSA were retrieved from the assemblies, and each gene was aligned using ClustalW 2.1 (*Larkin et al., 2007*). Maximum likelihood trees based on the concatenated sequences were reconstructed using RAxML 8.2.11 (*Stamatakis, 2014*) with GTRGAMMA model and 500 bootstrap replications.

## RESULTS

To compare the performance of sequencing platforms, the whole genome sequence of *V. aphrogenes* was obtained using different methods: Nanopore sequencing using MinION (Nanopore), PacBio sequencing using the SMRT system (PacBio), MP and PE reads from MiSeq (MiSeq-MP and MiSeq-PE) (Table S1). Using the Illumina corrected PacBio assembly (PacBio+Illumina) as a reference, overall accuracy and completeness of the assemblies were evaluated. As summarized in Table 1, PacBio or Nanopore alone was able to reconstruct two circular contigs without gaps, each corresponding to the two chromosomes of this bacterium. Furthermore, the Nanopore+Illumina hybrid assembly was highly consistent with the PacBio assembly. Indels derived from tandem repeats with different length (number of repeats) were major differences between the assemblies.
**Table 2  Genome assemblies of five Rumoiensis clade species.** The genomes of five species were reconstructed in two different methods, hybrid assembly and Nanopore-only assembly. Indels and mismatches in Nanopore-only assemblies were determined based on the differences from the hybrid assemblies. MinION read data shown here are those obtained after debarcoding and adaptor trimming with Porechop 0.2.2 (https://github.com/rrwick/Porechop).

| Strain | *V. algivorus* | *V. aphrogenes* | *V. casei* | *V. litoralis* | *V. rumoiensis* |
|---|---|---|---|---|---|
| Total reads | 172,016 | 42,584 | 332,715 | 163,459 | 192,869 |
| Total bases | 938,094,776 | 467,570,099 | 2,237,368,370 | 772,425,123 | 837,603,769 |
| Average read length (bp) | 5,454 | 10,980 | 6,725 | 4,726 | 4,343 |
| Number of contigs (reference) | 2 | 2 | 5 | 3 | 4 |
| Number of contigs (Nanopore) | 3 | 2 | 3 | 4 | 5 |
| Total length (bp) (reference) | 3,648,612 | 3,375,144 | 4,140,771 | 3,872,238 | 4,207,152 |
| Total length (bp) (Nanopore) | 3,711,100 | 3,371,144 | 4,118,045 | 3,920,009 | 4,326,255 |
| Indels per kb | 0.59 | 1.36 | 0.41 | 0.83 | 0.77 |
| Mismatches per kb | 0.38 | 0.05 | 0.03 | 0.32 | 0.06 |

**Table 3  G+C content stability determined from different assemblies.** G+C contents of the hybrid, Nanopore-only, and Illumina-only assemblies were calculated, respectively.

| Strain | Hybrid | Nanopore | Illumina |
|---|---|---|---|
| *V. algivorus* | 40.80 | 40.78 | 40.73 |
| *V. aphrogenes* | 42.13 | 42.17 | 42.06 |
| *V. casei* | 40.72 | 40.72 | 40.54 |
| *V. litoralis* | 42.01 | 41.96 | 41.94 |
| *V. rumoiensis* | 42.31 | 42.35 | 42.25 |

Reference genomes for *V. aphrogenes*, *V. algivorus*, *V. casei*, and *V. litoralis* were produced by hybrid approach using Unicycler. As Unicycler failed to reconstruct complete circular contigs for *V. rumoiensis*, Nanopore reads were assembled with Canu, and contigs were manually closed based on the assembly graph. The resulting circular contigs were polished with Illumina reads using Pilon. Using the Unicylcer hybrid or Canu+Pilon assemblies as the references, the error rates of Nanopore-only assemblies were estimated for each species. Consistent with previous reports (*Loman, Quick & Simpson, 2015*; *Wick et al., 2017a*), Nanopore-only assemblies had high per base error rates, with 0.4112–1.3644 indels per kb and 0.0265–0.3815 mismatches per kb (Table 2).

Despite the relatively high error rates in Nanopore-only asemblies, taxonomic measures were not highly affected by differences in sequencing or assembly methodologies. Overall, ANI calculated using different assemblies were highly consistent; the comparisons between the same pair of species showed a maximum difference of 0.42% (*V. algivorus* hybrid and *V. casei* Nanopore-only: 78.12%, *V. algivorus* Nanopore-only and *V. casei* Illumina: 77.70%) (Fig. 1). *In silico* DDH was more sensitive to the differences in methodology, with values ranging from 79.0 to 100 in the comparison of the same pair of strains (Fig. S1). A considerable difference was also not observed in G+C content, with a maximum difference of 0.04% between hybrid-Nanopore (*V. aphrogenes* and *V. rumoiensis*), 0.18% for hybrid-Nanopore and Illumina-Nanopore (*V. casei*) (Table 3).

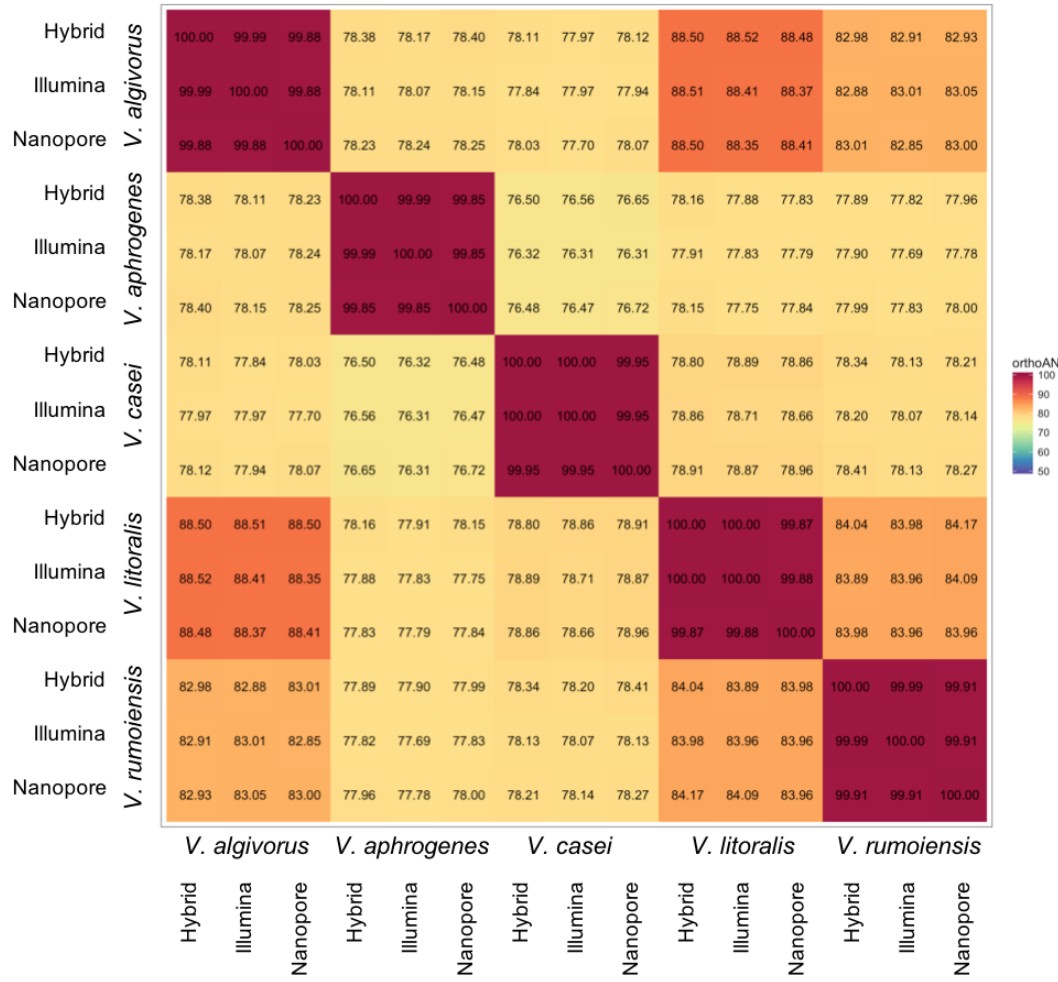

**Figure 1** **Heatmap representation of ANI values using different assemblies.** ANI values were calculated using Orthologous Average Nucleotide Identity Tool version 1.3 (*Lee et al., 2016*) and the values represented here are the orthoANI values.

To test the compatibility of Nanopore sequencing for species identification with MLSA, eight protein coding genes (*gapA*, *gyrB*, *ftsZ*, *mreB*, *pyrH*, *recA*, *rpoA*, and *topA*) used for *Vibrionaceae* MLSA were retrieved from the Nanopore-only assemblies of fives species and compared with those in the hybrid assemblies. As summarized in Table 4, each gene had zero to two deletions, and occasionally insertions or mismatches, in full length gene comparisons. The error frequencies were reduced if we compare 369–636 bp regions where original eight-gene MLSA used (*Tanaka et al., 2017*) (Table 4). Error frequencies against Sanger sequences were the same extent of those to Illumina or hybrid assemblies. Using MLSA gene sequences from the hybrid and Nanopore-only assemblies, phylogenetic trees were constructed, and the trees were compared based on their topology. Two trees were topologically identical (symmetric difference of 0) (Fig. S2) with only one node showing differing bootstrap support value.

**Table 4  Evaluation of protein coding gene sequences for MLSA retrieved from Nanopore-only assembly.** Gene sequences from Nanopore-only assemblies were retrieved and compared with the genes from the hybrid assemblies.

| Error type | MLSA genes | | | | | | | |
|---|---|---|---|---|---|---|---|---|
| | *ftsZ* | *gapA* | *gyrB* | *mreB* | *pyrH* | *recA* | *rpoA* | *topA* |
| **V. algivorus** | | | | | | | | |
| Length (bp) | 1,218/435 | 996/636 | 2,439/588 | 1,044/507 | 735/369 | 1,047/486 | 993/378 | 2,628/420 |
| Mismatch | 0/0 | 0/0 | 0/0 | 0/0 | 0/0 | 0/0 | 2/0 | 2/0 |
| Insertion | 0/0 | 0/0 | 0/0 | 0/0 | 0/0 | 0/0 | 1/0 | 2/0 |
| Deletion | 0/0 | 0/0 | 0/0 | 2/0 | 1/0 | 0/0 | 1/0 | 1/0 |
| **V. aphrogenes** | | | | | | | | |
| Length (bp) | 1,215/435 | 996/636 | 2,439/588 | 1,044/507 | 735/369 | 1,047/486 | 993/378 | 2,628/420 |
| Mismatch | 0/0 | 0/0 | 0/0 | 0/0 | 0/0 | 0/0 | 0/0 | 0/0 |
| Insertion | 0/0 | 0/0 | 0/0 | 0/0 | 0/0 | 0/0 | 0/0 | 0/0 |
| Deletion | 2/2 | 0/0 | 2/2 | 1/0 | 1/0 | 0/0 | 0/0 | 2/0 |
| **V. casei** | | | | | | | | |
| Length (bp) | 1,218/435 | 996/636 | 2,439/588 | 1,044/507 | 732/369 | 1,041/486 | 993/378 | 2,628/420 |
| Mismatch | 0/0 | 0/0 | 0/0 | 0/0 | 0/0 | 0/0 | 0/0 | 0/0 |
| Insertion | 0/0 | 0/0 | 0/0 | 0/0 | 0/0 | 0/0 | 0/0 | 0/0 |
| Deletion | 2/0 | 0/0 | 2/0 | 1/0 | 1/0 | 0/0 | 0/0 | 2/0 |
| **V. litoralis** | | | | | | | | |
| Length (bp) | 1,218/435 | 996/636 | 2,439/588 | 1,044/507 | 735/369 | 1,047/486 | 993/378 | 2,628/420 |
| Mismatch | 0/0 | 0/0 | 0/0 | 0/0 | 1/0 | 0/0 | 0/0 | 0/0 |
| Insertion | 0/0 | 0/0 | 1/0 | 0/0 | 1/0 | 0/0 | 0/0 | 0/0 |
| Deletion | 0/0 | 0/0 | 0/0 | 0/0 | 1/0 | 0/0 | 0/0 | 0/0 |
| **V. rumoiensis** | | | | | | | | |
| Length (bp) | 1,215/435 | 996/636 | 2,439/588 | 1,044/507 | 735/369 | 1,047/486 | 993/378 | 2,628/420 |
| Mismatch | 0/0 | 0/0 | 0/0 | 0/0 | 0/0 | 0/0 | 0/0 | 0/0 |
| Insertion | 0/0 | 0/0 | 0/0 | 0/0 | 0/0 | 0/0 | 0/0 | 1/0 |
| Deletion | 0/0 | 1/1 | 0/0 | 0/0 | 0/0 | 0/0 | 0/0 | 0/0 |

**Notes.**
Full length/MLSA region.

# DISCUSSION

Comprehensive comparative genomics is one of the most promising methodologies in establishing reproducible and reliable criteria toward the next generation microbial taxonomy (*Chun et al., 2018*). Accelerating use of the genome-based taxonomy increased the demand of fast, high quality, and cost-effective genome sequencing, and the ability to produce long reads in relatively short time, including library preparation, make Nanopore sequencing more attractive. Our data evaluating three major OGRI involving ANI, *in silco* DDH, and G+C content for the genome-based taxonomy using the phylogenetically robust and genomically distinct Rumoiensis clade species in *Vibrionaceae* suggests that genome sequences obtained using the ONT MinION are available for genome-based microbial taxonomy.

For Nanopore sequencing of five *Vibrio* species, the barcoding kit was used to increase the cost-effectiveness. Using two barcodes per species, the sequence data obtained for
each species ranged from 0.47 Gb to 2.2 Gb, and the average lengths were between 4.3 kb to 11.0 kb (Table 2). As MinION could typically generate 5–10 Gb DNA in a single run (*Magi et al., 2017*), the data amounts for each genome obtained in this study are in the range of typically reported values. Combining this with Illumina reads, the Nanopore-based hybrid assembly successfully reconstructed the gap-closed, finished-grade circular genomes involving two major chromosomes commonly possessed by *Vibrionaceae* species. Ability to reconstruct the complete genomes without the other highly intensive works has great potentials to enhance not only the genome-based taxonomy but also acquisition of the complete genomes for other members of *Vibrionaceae*. Complete genomes could reinforce our knowledge on the genome plasticity, one of the major topics in elucidating *Vibrio* biodiversity, pathogenesis, and evolution (*Gomez-Gil et al., 2014*).

MLSA is a powerful method for inferring the evolutionary history of particular taxonomic groups. In *Vibrionaceae* systematics, MLSA is particularly important as *Vibrionaceae* species cannot be identified based on a single molecular marker such as 16S rRNA gene due to the low discriminatory power (*Sawabe, Kita-Tsukamoto & Thompson, 2007*; *Gomez-Gil et al., 2014*). As one of the disadvantages of the Nanopore sequencing is the higher error rates compared to Illumina or PacBio sequencing, it is worth evaluating whether these error rates significantly affect the MLSA or not. Unexpectedly, however, low frequencies of mismatch and indel were observed in eight genes typically used for MLSA designed for *Vibrionaceae* taxonomy (Table 4). In the broad phylogenetic network reconstruction that is generally performed in the initial step of species and/or clade identification of *Vibrionaceae*, each Rumoiensis clade species forms a robust cluster on each terminal node even if we use gene sequences retrieved from Nanopore-only assemblies. We further conclude that genes from the Nanopore-only assemblies were able to reconstruct the MLSA phylogeny of the Rumoiensis clade species.

Comparison of the Rumoiensis clade species showed Nanopore-only assembly can be utilized to discriminate between species based on OGRI. Additionally, preliminary comparisons of genomes of two strains in the Ponticus clade and three strains in the Splendidus clade of the genus *Vibrio* sharing 99.9% and 99.8%–99.9% 16S rRNA gene identity, respectively, sequenced using MinION shows 98.7% ANI between Ponticus clade strains and 95.4%–97.0% between Splendidus clade strains, indicating these strains belong to the same species. While this suggests availability of Nanopore sequencing for comparison of closely related strains, caution needs to be taken as high error rates may obstruct the applications such as population genetics and SNP detection. Nonetheless, this fast, reliable and cost-effective means of whole genome sequencing has the potential to advance genome-based taxonomy and development of automated taxonomy solely based on the genomic data.

## CONCLUSIONS

The complete genomes of five closely-related vibrios were reconstructed using Nanopore sequencing technology. Although Nanopore-only assemblies have previously been described as not being suitable for sequence/allele typing or small variant studies due

to high error rates (*Magi et al., 2017*; *Wick et al., 2017a*), our dataset shows that Nanopore-only assemblies can be used to discriminate species based on whole genome similarity for taxonomic purposes.

## ACKNOWLEDGEMENTS

We gratefully thank Dr. Toshiyoshi Araki at Iga Research Institute of Mie University and Dr. Isao Yumoto at National Institute of Advanced Industrial Science and Technology for providing bacterial strains. We also thank to Prof. Masahira Hattori and Dr. Wataru Suda at the Tokyo University for sequencing parts of *Vibrio* genomes using Illumina.

### Funding

This work was partly supported by Kaken (16H04976), and internal University budgets from "Subsidies for operating expenses". There was no additional external funding received for this study. The funders had no role in study design, data collection and analysis, decision to publish, or preparation of the manuscript.

### Grant Disclosures

The following grant information was disclosed by the authors:
Kaken: 16H04976.
Internal University budgets from "Subsidies for operating expenses".

### Competing Interests

The authors declare there are no competing interests.

### Author Contributions

- Mami Tanaka conceived and designed the experiments, performed the experiments, analyzed the data, contributed reagents/materials/analysis tools, prepared figures and/or tables, authored or reviewed drafts of the paper, approved the final draft.
- Sayaka Mino and Yoshitoshi Ogura performed the experiments, contributed reagents/materials/analysis tools, authored or reviewed drafts of the paper, approved the final draft.
- Tetsuya Hayashi contributed reagents/materials/analysis tools, authored or reviewed drafts of the paper, approved the final draft.
- Tomoo Sawabe conceived and designed the experiments, contributed reagents/materials/analysis tools, prepared figures and/or tables, authored or reviewed drafts of the paper, approved the final draft.

### Data Availability

Raw data and assembled genomes are deposited in DDBJ/EMBL/GenBank under BioProject PRJDB5783 involving BioSamples SAMD00080445, SAMD00057777, SAMD00080446, SAMD00116203, and SAMD00080447 for *V. algivorus*, *V. aphrogenes*, *V. casei*, *V. litoralis*, and *V. rumoiensis*, respectively.

## Supplemental Information

Supplemental information for this article can be found online at http://dx.doi.org/10.7717/peerj.5018#supplemental-information.

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
