# Peer review of "Availability of Nanopore sequences in the genome taxonomy for Vibrionaceae systematics: Rumoiensis clade species as a test case"

_PeerJ, doi:10.7717/peerj.5018_

## Round 0.1 · original submission · Major Revisions

Dear Dr. Sawabe,

Please address all suggestions and changes of the reviewers comments, especially suggestions of reviewer 2, and provide a point by point response letter.

Thank you in advance.

·

Basic reporting

The manuscript is well written and structured. The background provided is correct as well as the references employed. Tablas and Figures are adequate.

Experimental design

The aims and objectives are clear, as is the rationale of the experimental design.
The methods are in general described with enough detail. Some minor points or doubts are: 1) why the ZoBell 2216E broth was prepared with natural seawater? That means that the salt concentration was doubled?. 2) More detail is needed within the heading "Phylogenetic reconstruction and tree comparison"; it is a MLSA? Which aim?. 3) It is no needed to define the acronyms more than once.
The investigation carried out was carried out to a high standard, and the comparison of the differente NGS methods is rigurous and complete.

Validity of the findings

The data presented are robust and the conclusion well based on the findings. The validity of nanopore-only assemblies for taxonomic purposes may have an impact in this field within Microbiology.

Additional comments

The manusctipt constitutes an interesting piece of work, than may have an impact in microbial taxonomy.

Reviewer 2 ·

Basic reporting

There is no reason to believe nanopore provides error free sequences. And some previous studies have pointed out error rates>30%! Although the authors mentioned in their paper that nanopore is highly accurate and make very very few reading errors, I am not convinced. The authors list in Table 4 mismatches due to nanopore errors, the reader really needs more details about the %error, types of most frequent errors (A, T, G, C?), comparison with previsou Sanger sequences (eg MLSA), which the authors have abundant databases(eg Sawabe et al. 2007; JBac).

Experimental design

include Sanger sequences in the comparison.

Validity of the findings

requires evaluation.

Additional comments

I have major doubts about the validity and reliability of nanopore for taxonomic and ecogenomic applications. The authors failed to properly prove the relevance of nanopore in this version.

Reviewer 3 ·

Basic reporting

This a well-written manuscript, The rational of this work is convincingly spelled out, background is sufficiently defined, relevant illustrative material to support the authors' statements is provided.

Experimental design

The evaluation of advanced genome sequencing technique is timely. In the context of this manuscript I would suggest to clearly define the aims of this work in the last paragraph of the Introduction.

Validity of the findings

Indeed, the novelty aspect should be highlighted. In my view the manuscript will benefit from comparative analysis of advantages and disadvantages of the three current approaches for whole genome sequencing. This could be useful and could be included in the Abstract.
It appeared that in silico DDH values are not matching. Since this is one of most important characteristics for species discrimination, the authors should highlight this outcome, comment on the possible reasons and perhaps future work to overcome this drawback.

Additional comments

Please revise Table 2 by minor re-arrangement, e.g., removing the two rows with same words. Consider to improving Table 4 as well.

Please provide the source of the type strains used for the sequencing.Since some data retrieved from previous sequencing, the authors should double check if the same type strains obtained from the same sources have been used. These maybe the reason for some discrepancies in sequencing results.

---

## Round 0.2 · accepted · Accept

Dear Mami and Tomoo,

You have responded the questions and included the suggestions of the reviewers and so I have no more questions. It is a pleasure to inform you that your manuscript has been Accepted for publication. Congratulations!

# Reviewer 2 ·

Basic reporting

The authors have responded to my main concern and so I have no more questions. Nanopore can be indeed an excellent tool for vibrio taxonomy as demonstrated by the authors in this test case of rumoensis.

Experimental design

ok.

Validity of the findings

ok

Additional comments

ok